# Foreground Harmonization and Shadow Generation for Composite Image

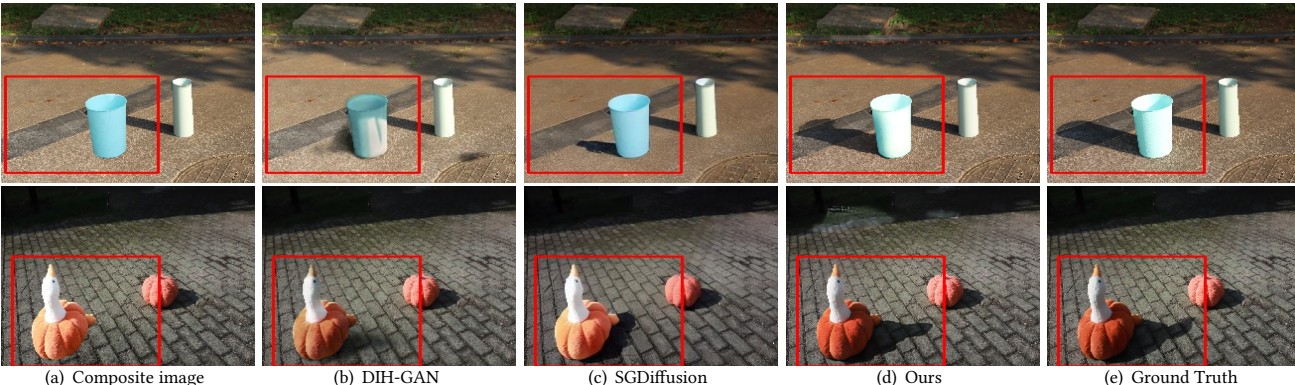

(a) Composite image     (b) DIH-GAN     (c) SGDiffusion     (d) Ours     (e) Ground Truth

**Figure 1: Illumination editing effects. A composite image as input, it is possible to generate a harmonious foreground image as well as realistic cast shadows. From left to right are composite image, results of DIH-GAN, SGDiffusion and our method, Ground Truth, respectively.**

## ABSTRACT

We propose a method for light and shadow editing of outdoor disharmonious composite images, including foreground harmonization and cast shadow generation. Most existing work can only perform foreground appearance editing tasks or only focus on shadow generation. In fact, lighting not only affects the brightness and color of objects, but also produces corresponding cast shadows. In recent years, diffusion models have demonstrated their strong generative capabilities, and due to their iterative denoising properties, they have a significant advantage in image restoration tasks. But it fails to preserve content structure of image. In this purpose, we propose an effective model to tackle the problem of foreground light-shadow editing. Specifically, we use a coarse shadow prediction module (SP) to generate coarse shadows for foreground objects. Then, we use the predicted results as prior knowledge to guide the generation of harmony diffusion model. In this process, the primary task is to learn lighting variation to harmonize foreground regions. The secondary task is to generate high-quality cast shadow containing more details. Considering that existing datasets do not support the dual tasks of image harmonization and shadow generation, we construct a real outdoor dataset, IH-SG, covering various lighting conditions. Extensive experiments conducted on existing benchmark datasets and the IH-SG dataset demonstrate the superiority of our method.

## KEYWORDS

Image harmonization, shadow generation, diffusion model

## 1 INTRODUCTION

Image coposite refers to the process of combining images from different sources to create new images, which has a variety of applications like advertisement propaganda and digital entertainment. However, due to variations in lighting conditions, camera parameters, and other factors, composite images often have inconsistent lighting statistics compared to real images. This necessitates image harmonization to adjust the appearance of the foreground for visual consistency. Additionally, most existing image harmonization methods focus solely on the lighting effects of foreground regions. However, lighting also produces corresponding shadow effects. These shadows provide important clues about object shape, position, and relative depth, conveying information about volume and depth to observers. Therefore, shadow generation is equally essential for achieving lighting-shadow consistency.

For image harmonization, most traditional methods focus on better matching low-level appearance statistics. They mainly focus on color correction and lighting compensation [3, 31, 33, 49] for the foreground area, or adjustments based on gradient information [18, 30, 45] and multiscale statistics [43] of foreground and background regions. They fail to solve the significant appearance differences between foreground and background images. Deep learning based methods provide powerful capabilities for modeling regional

appearances to facilitate harmonization. Some methods [7, 47] explore the semantic information of synthesized images to reconstruct coordinated images. Several methods [4, 6, 14, 23] have been proposed to explore domain adaptation, aiming to bring the predicted foreground closer to the original background domain. Guo et al. [13] introduced the Retinex theory into the image harmonization task. These methods [11, 12] explored Transformer-based image harmonization. Considering the limitations of existing datasets, self-supervised or semi-supervised methods proposed [19, 27, 29, 48]. CDTNet [5] and PCT-Net [9] adjust the image through color transformation, but ignore lighting effects. Tan et al. [44] used unrelated L, a, b features to guide image reconstruction.

For shadow generation, render-based methods [39–41] require explicit knowledge of lighting, reflectance, material properties, and scene geometry to generate shadows for inserted virtual objects using rendering techniques. However, obtaining such knowledge is often impractical in real-world scenarios. The estimated results are influenced by the accuracy of the input information [8, 17]. Deep learing-based methods [16, 25], on the other hand, directly learn the mapping from input images without foreground shadows to output images with foreground shadows, without the need for explicit knowledge of lighting, reflectance, etc. Bao et al. [1] consider the harmonized appearance and illumination of the foreground objects while generating reasonable shadows for the foreground objects. But this method only studied indoor images.

To address these issues, we propose a novel method for both image harmonization and shadow generation in this paper. Considering the powerful generation capability of diffusion models, inspired by [10], we use a condition diffusion model as the backbone network. Recognizing that images provide rich structural and semantic features to assist image reconstruction compared to textual information. We use composite image with coarse shadows as condition to guide the diffusion model. Since our task also requires generating plausible cast shadows for the foreground objects, we introduce a feature extraction module, a channel-spatial cross attention and a shadow generation model to generate coarse shadows. The main task of the harmony diffusion model is to learn the lighting changes in the background to handle the foreground area harmoniously, and refine the shadow area to generate high-quality shadows.

The existing dataset are not well-suited for our task. IHarmony4 [6] provides different color conversions but lacks attention to lighting. RealHM [19] and RdHarmony [2] require a significant amount of manpower and technical resources. CcHarmony [29] focuses on realistic lighting changes, but it requires high requirements for the shooting process. ShadowAR [25] dataset is collected through rendering models. However, the attributes of shadows may also not match those of real images. DESOBA and DESOBAv2 [16, 26] use real images as target images to remove shadows from the foreground to generate composite images. Bao et al. [1] proposed an indoor dataset for foreground harmonization and shadow generation, but only focusing on indoor scenes. So we construct a new outdoor real-world dataset (IH-SG) for image harmonization and shadow generation task. We obtain composite images through re-lighting foreground objects.

Our contributions can be summarized as follows:

- We constructe a new outdoor real-world dataset (IH-SG) for image harmonization and shadow generation task.
- We propose a new image light-shadow editing method based on condition diffusion model, which can achieve controllable harmonization of foreground regions and reasonable generation of cast shadows.

Extensive experiments conducted on both public datasets and our IH-SG dataset demonstrate the effectiveness of our method.

## 2 RELATED WORK

Our task aims to handle the illumination of the foreground regions and generate reasonable cast shadows for foreground objects, while image harmonization or shadow generation tasks can only solve one of them.

### 2.1 Image Harmonization

Traditional image harmonization methods primarily focus on adjusting the low-level appearance statistics between foreground objects and the background, such as color statistics [3, 31, 33, 49], and gradient information [18, 30, 45]. The limited representation capability of low-level features can negatively impact their performance. Especially when there are significant differences between the foreground and background regions, these methods often struggle to produce satisfactory results.

Recent research has built reasonably sized datasets [6, 19, 29], to advance learning-based approaches. CNN-based methods analyze semantic information [7, 47]. Since image harmonization adjusts the foreground lighting or style to match the background, domain adaptation methods [4, 6, 23] have also been proposed to explore the idea of domain harmonization. Guo et al. [13] introduced the Retinex theory into the image harmonization task and proposed decomposing the synthetic image into reflectance and illumination. With the rise of Transformers, Guo et al. [11, 12] applied the Transformer framework to image harmonization tasks. But intrinsic decomposition is a difficult problem. Some methods treat image harmonization as a style transfer problem. These methods have achieved advanced research results through contrastive learning [14], high resolution [20] or color space adjustment [5, 9, 44]. Shen et al. [38] trained Global Perception Adaptive Coordination Kernel. Bao et al. [1] utilized a multi-scale attention mechanism and illumination exchange strategy to harmonize objects and generate cast shadow. However, these methods for indoor scenes are difficult to generalize to outdoor scenes. Unlike existing methods, we learn the illumination of images through diffusion model to generate the consistent illumination of the foreground object itself with the background, as well as corresponding cast shadow.

### 2.2 Shadow Generation

The existing work on shadow generation can be divided into two categories: rendering-based methods and image-to-image translation methods. Rendering-based methods require explicit knowledge of lighting, reflectance, and scene geometry to generate shadows for inserted virtual objects using rendering techniques. However, such detailed knowledge relies on user input [21, 24] or model prediction [8, 22]. Sheng et al. [40] explored the generation of controllable soft shadows. And then, they introduced the concept of

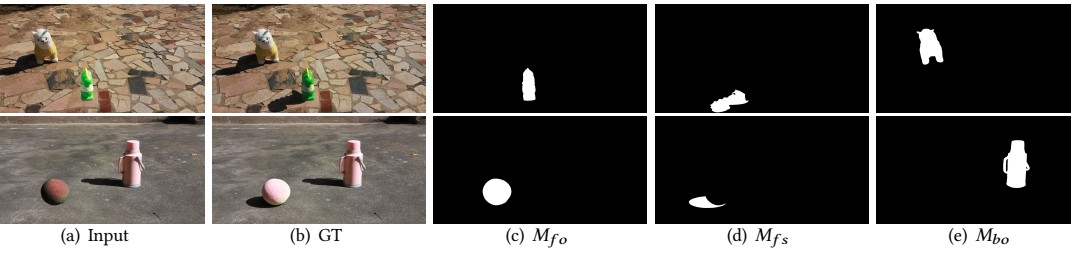

| (a) Input | (b) GT | (c) $M_{fo}$ | (d) $M_{fs}$ | (e) $M_{bo}$ | (f) $M_{bs}$ |

**Figure 2: IH-SG dataset. From left to right, there are composite images, real images, foreground object masks, foreground shadow masks, background object masks and background shadow masks, respectively.**

pixel height [39, 41] and explored the correlation between objects, ground, and camera poses. In the absence of user interaction, Gardner [8] attempted to recover explicit lighting conditions and scene geometry based on a single image, but inaccurate estimates may lead to unsatisfactory results.

Image-to-image translation methods learn the mapping from input images without foreground shadows to output images with foreground shadows, without requiring explicit knowledge of lighting, reflectance, etc. Hu et al. [17] proposed a method that can adapt to different scenarios, but failed to generate shadows in complex scenes. For instance, ShadowGAN [52] utilized both global and local conditional discriminator to enhance the realism of generated shadows. Liu et al. [25] released the ShadowAR dataset and proposed an attention-guided network for shadow generation. Yan et al. [16] addressed real-world scenes and generated plausible shadows. SGDiffusion [26] focused on the shadow generation problem based on a diffusion model. DMASNet [46] decomposed shadow mask prediction into box prediction and shape prediction. However, the shadows generated by these methods are still not accurate enough.

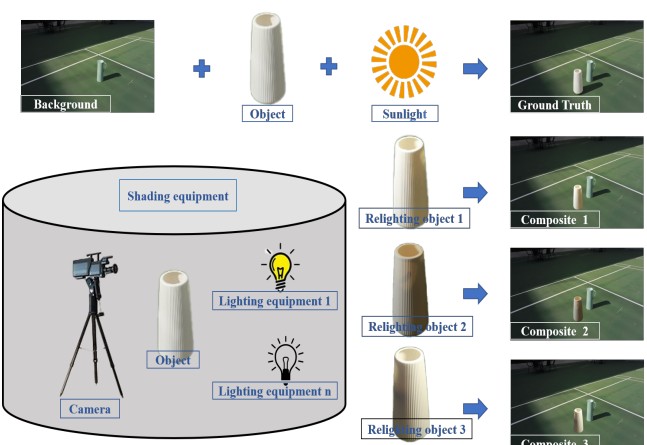

**Figure 3: The pipeline of dataset construction. Take background images and real images under natural sunlight. Block and relight the foreground object. Finally, obtain the composite image.**

## 2.3 Diffusion Model

Diffusion-based generative models recently produced amazing results with improvements adopted in denoising diffusion probabilistic models [15], which becomes increasingly influential in the field of low-level vision tasks, such as superresolution [37], inpainting [28], and colorization [36]. The methods [32, 34, 35] explored different modal conditions into the diffusion process, achieving controllability of the generated content of the generative model. Pallette [36] was proposed as a general image-to-image framework to solve the image restoration with conditional denoising diffusion probability models. The methods [50, 51] were proposed to generate results towards expectations. However, most of these methods focus on synthetic degradation, such as image coloring, image restoration, and super-resolution. In this article, we explore the problem of foreground harmony and shadow generation in the real world with limited training pairs. We build our model upon shadowdiffusion [10] to address the above issues.

## 3 IH-SG DATASET

Lighting not only results in different color brightnesses of foreground objects but also leads to the generation of corresponding cast shadows. In this article, we simultaneously focus on the harmony of foreground objects and the generation of corresponding realistic shadows. Therefore, we have constructed a high-quality real outdoor dataset IH-SG, including composite images $I_c$, real images $I_{real}$, foreground obeject masks $M_{fo}$, foreground shadow masks $M_{fs}$, background object masks $M_{bo}$ and background shadow masks $M_{bs}$.

### 3.1 Image Collection

We take photos outdoors that meet our requirements, including background images, real images, and relighting images:

**Background images** $I_{back}$**:** Choose optimal weather conditions and time periods with moderate lighting, avoiding excessively dim or harsh lighting conditions. Additionally, steer clear of rainy days or periods around sunrise or sunset, as these times exhibit significant changes in lighting. Select appropriate shooting angles and positions, stabilize the camera on a tripod, and control it via a mobile device to ensure camera stability.

**Real images** $I_{real}$**:** Without altering camera parameters, positions, etc., placing foreground objects and capturing images to be used as real images in the training set. The camera remains stable

**Figure 4: Pipeline of the proposed method. IH-SG Diffusion model includes coarse shadow prediction network (SP) and denoise network $f_t$. Given a disharmonious image, our model can generate a harmonious image with controllable foreground objects and reasonable cast shadows.**

throughout the process, ensuring consistency between background and real images, thus reducing alignment and correction efforts during subsequent image synthesis. Rapid placement of foreground objects generally assumes minimal changes in lighting between background and real images captured in a short time.

**Relighting foreground images $I^i_{relight}$:** Without altering the shooting scene or camera position, we employ appropriately sized and shaped shading equipment to shade the scene, ensuring that neither the camera nor the objects are affected. Shading equipment is utilized to effectively block external light interference. Placing lights and adjusting their brightness and direction to illuminate foreground objects. Camera parameters such as exposure time, aperture size, and ISO sensitivity are adjusted based on the actual shooting environment and lighting conditions to achieve the desired exposure effects. Subsequently, adjust the lighting conditions to capture different relighted images $I^i_{relight}$, $i \in N$, where N represents different lighting conditions.

### 3.2 Image Synthesis

Based on the acquired real images $I_{real}$, background images $I_{back}$, and relighting images $I^i_{relight}$, composite images $I_c$ are obtained. To obtain refined data, we used Photoshop to obtain corresponding masks, including foreground object masks $M_{fo}$, foreground shadow masks $M_{fs}$, background object masks $M_{bo}$, and background shadow masks $M_{bs}$. Then, we use the obtained foreground object masks to extract relighting foreground objects and merge them with background images to generate composite images:

$$I_c = I_{relight} \times M_{fo} + I_{back} \times \left(1 - M_{fo}\right). \tag{1}$$

Then, $I_c$ and $I_{real}$ form a pair of input composite image and ground-truth target image. Due to shooting conditions, there may be significant differences between the background images and the real images in the background area. If there are differences in color or brightness, they could be adjusted using style transfer techniques. Additionally, some unsuitable images could be filtered out. After that, we obtained tuples in the form of $\{I_c, M_{fo}, M_{fs}, M_{bo}, M_{bs}, I_{real}\}$, which will be used for model training.

## 4 METHOD

### 4.1 Problem Definition

The input is a tuple $(I_c, M_{fo})$, where $I_c \in R^{H \times W \times C}$, with $H$ and $W$ representing the height and width of image, and $M_{fo} \in R^{H \times W \times 1}$. This model aims to generate foreground object that is consistent with the background, and to generate reasonable cast shadow.

### 4.2 Coarse Shadow Predict Module

The coarse shadow prediction module aims to predict cast shadows for foreground objects, including a background feature extraction network (BE) and a shadow generation network (SG). The composite image and foreground object mask are the inputs.

*4.2.1* **Background Extraction Module**. For the shadow generation task, while complete background information may provide more details, it does not directly yield reasonable shadows for image-to-image transformation networks. This is because it may not adequately focus on crucial areas in the background, such as objects and their shadow information. Therefore, the BE module can learn relevant information from the background image to generate attention maps for reference objects and their shadows.

The module adopts an encoder-decoder network with an attention mechanism as basic architecture, comprising an encoder and two decoders. The composite image without foreground object shadow andforeground object mask are concatenated along the channel dimension and serve as input to the encoder $E$. The extracted high-level features are fed into two separate branches of decoders. One decoder $D_1$ predicts the reference object mask $M_{bo}$, while the other decoder $D_2$ predicts the corresponding shadow mask $M_{bs}$:

$$M_{bo} = D_1(E(I_c)), \qquad (2)$$

$$M_{bs} = D_2(E(I_c)). \qquad (3)$$

*4.2.2* **Shadow Generation Module**. Given a composite image $I_c$ without foreground shadow and a foreground object mask $M_{fo}$, this module aims to generate coarse foreground shadow $I_{shadow}$. The specific network structure is as follows: Two same encoders, one decoder, and one special channel-spatial cross-attention mechanism (CSCA).

Through the BE module, we can identify key areas in the background image that are beneficial for shadow generation. Inspired by [16], in order to better utilize the information in the background, we adopt foreground encoder $E_F$ and background encoder $E_B$, respectively. The foreground encoder $E_F$ takes the concatenation of the composite image $I_c$ and the foreground object mask $M_{fo}$ as input, generating a foreground feature map $X_f$. The background encoder $E_B$ takes the concatenation of $I_c$ and $M_{bos}$ as input to generate a background feature map $x_b$:

$$X_f = E_F(I_c, M_{fo}), \qquad (4)$$

$$X_b = E_b(I_c, (M_{bo} + M_{bs})). \qquad (5)$$

Taking inspiration from existing attention methods, we introduce a channel-spatial cross-attention (CSCA) to assist the foreground feature map $X_f$ in obtaining relevant reference information from the background feature map $X_b$. Then, the decoder D is used to predict coarse shadow images for foreground objects:

$$I_{shadow}, M_{shadow} = D(CSCA(X_f, X_b)). \qquad (6)$$

*4.2.3* **Channel-Spatial Cross-Attention Module**. Obtaining relevant illumination information is crucial for generating accurate foreground shadows. Inspired by previous attention-based methods, we used a Channel-Spatial Attention Module (CSAM), as shown inFigure 5, to help the foreground feature map $X_f$ extract relevant illumination information from the background feature map $X_b$. By constructing the relative positional relationship between reference information and foreground through this module, it effectively guides the generation of foreground shadows in a reasonable direction.

**Channel cross-attention:** To project foreground and background features into a common space, we reshape $X_f \in R^{W \times H \times C}$

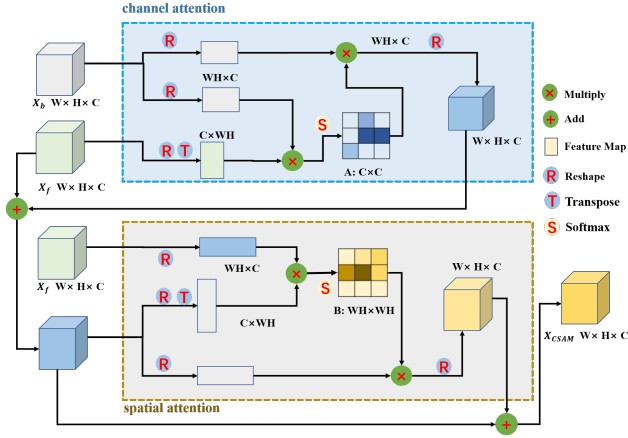

**Figure 5: Channel-Apatial Cross-Attention Module. It includs channel cross-attention and spatial cross-attention submodules.**

.

to $X_f^r \in R^{WH \times C}$ and $X_b \in R^{W \times H \times C}$ to $X_b^r \in R^{WH \times C}$. Then, we compute the dependencies between any two elements of $X_f$ and $X_b$ in the global context:

$$A = softmax\left((X_f^r)^T X_b^r\right). \qquad (7)$$

Using the obtained similarity map A, we incorporate information from $X_f^r$, then reshape it, and obtain the weighted feature map $X_{b2}$:

$$X_{b2} = X_f + reshape\left(X_b^r A\right). \qquad (8)$$

**Spatial cross-attention:** Similar to the channel cross-attention, we reshape $X_f \in R^{W \times H \times C}$ to $X_f^r \in R^{WH \times C}$ and $X_{b2} \in R^{W \times H \times C}$ to $X_{b2}^r \in R^{WH \times C}$. Then compute the similarity between feature maps:

$$B = softmax\left(X_f^r(X_{b2}^r)^T\right). \qquad (9)$$

Using the obtained similarity image B, weight $X_{b2}^r$, then reshape it to obtain the weighted feature map $X_{CSAM}$:

$$X_{CSAM} = X_{b2} + reshape\left(BX_{b2}^r\right). \qquad (10)$$

## 4.3 Harmony Diffusion Module

Controlling the generation of desired images in a controllable manner poses a challenging task for diffusion models. Especially when the objective is to obtain harmonious foreground images, it is crucial to ensure that the foreground and background share the same lighting distribution while preserving the content and structural information of the foreground objects. With the introduction of CLIP technology, text-guided diffusion models offer some controllable guidance. However, we recognize that images often provide more information than long texts. Therefore, in this module, we use compsite images with coarse shadows as conditions to guide the controllable generation of the diffusion model.

Diffusion model generates an image $x_0$ by denoising a random image following a Gaussian distribution $x_T \sim \mathcal{N}(0, I)$ progressively closer to the data distribution through multiple denoising steps

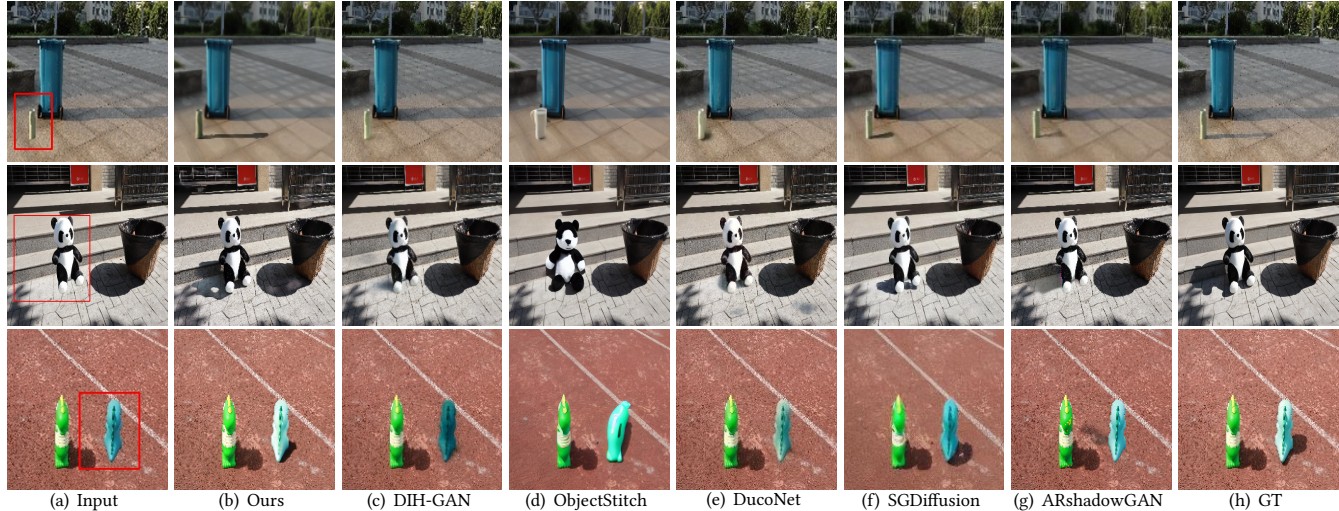

|          |          |             |                |             |                |               |         |
| :------: | :------: | :---------: | :------------: | :---------: | :------------: | :-----------: | :-----: |
| (a) Input | (b) Ours | (c) DIH-GAN | (d) ObjectStitch | (e) DucoNet | (f) SGDiffusion | (g) ARshadowGAN | (h) GT |

**Figure 6: Three testing cases of diferent methods on IH-SG dataset. From left to right are composite images, the results of our results, DIH-GAN [1], ObjectStitch [42], DucoNet [44] and the SGDiffusion [26], ARshadowGAN [25] and ground truth, respectively.**

$x_{T-1}, ..., x_0$. Diffusion model is divided into forward diffusion and inverse denoising phases.

**Forward process.** To construct training data, the forward process involves adding noise perturbations to the training image $x_0$ to generate noisy data $x_1, ..., x_T$:

$$x_t = \sqrt{\bar{\alpha}_t}x_0 + (1 - \bar{\alpha}_t)\epsilon, \tag{11}$$

with $\epsilon \sim \mathcal{N}(0, I)$, and $\bar{\alpha}_t = \prod_{s=0}^{t}\alpha_s = \prod_{s=0}^{t}(1 - \beta_s)$.

**Reverse process.** The reverse process aims to derive the posterior distribution for the less noisy image $x_{t-1}$ given the more noisy image $x_t$ using the denoising network $f_\theta$:

$$p(x_{t-1}|x_t, x_0) \sim \mathcal{N}(x_{t-1}; \mu_t(x_0, x_t), \sigma_t^2 I). \tag{12}$$

In addition to adjusting the lighting effect of foreground objects, there are also issues with predicting rough shadows in the previous stage, which require further refinement through the network. We have observed the following issues:

- The shape of the shadows generated in the previous stage is unrealistic.
- The lighting of the foreground is inconsistent with the background image.

We propose a method that uses composite images with coarse foreground object shadow (y) as conditional guidance to generate harmonized foreground object with realistic cast shadow. We train the denoising network $f_\theta$ to predict $x_0$ instead of the noise $\epsilon$:

$$x_0, m_{t-1} = f_\theta(x_t, y, m, t). \tag{13}$$

Following [15], our harmony diffusion model objective function is

$$\mathcal{L}_{pix} = \|x_{gt} - x_0\|_2^2, \tag{14}$$

where $x_{gt}$ is the real image, m is the predicted foreground object-shadow mask. Considering that we need to iteratively optimize the shadow area, we also need to calculate the loss between the

foreground object-shadow mask $m_f$ and the generated foregroung object-shadow mask $m_t$ for our method :

$$\mathcal{L}_{mask} = \|m_f - m_t\|_2^2. \tag{15}$$

Therefore, the total loss can be formulated as:

$$\mathcal{L}_{Total} = L_{pix} + 0.2 \times L_{mask}. \tag{16}$$

## 5 EXPERIMENTS

### 5.1 Experimental Setups

The proposed method is implemented using PyTorch, and training is performed using two GeForce RTX 3090. The training epoch is set to 1000. We utilize the Adam optimizer with a momentum of (0.9, 0.999). The initial learning rate is set to 0.9. Following, we employ the Kaiming initialization technique to initialize the weights of the proposed model, and use a 0.9999 exponential moving average (EMA) throughout all experiments. We adopt a U-Net architecture similar to the denoiser $\epsilon_\theta$ in [10]. Training is carried out with 200 diffusion steps T and a noise schedule $\beta_t$ that linearly increases from 0.0001 to 0.02, and inference is performed with 200 steps.

### 5.2 Dataset and Evaluation Metrics

We evaluated the performance of our method on IH-SG for object harmonization and shadow generation tasks. We resized the input and ground truth images to a size of 256 × 256 pixels. We calculated the Root Mean Square Error (RMSE), the Structural Similarity Index (SSIM) , fMSE, fSSIM for the generated images. And fMSE (resp., fSSIM) means MSE (resp., SSIM) within the foreground regions. In general, smaller values of RMSE and fMSE and larger values of SSIM and fSSIM indicate better quality of the generated images.

### 5.3 Comparison with Baselines

We compare with following methods: DIH-GAN [1], ObjectStitch [42], DucoNet [44], SGDiffusion [26] and ARshadowGAN [25].

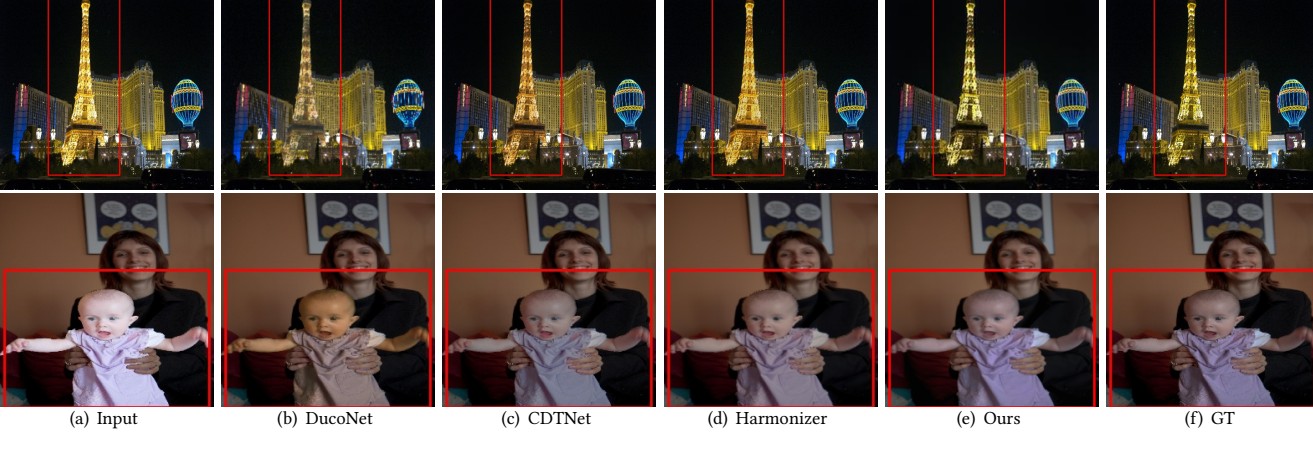

| (a) Input | (b) DucoNet | (c) CDTNet | (d) Harmonizer | (e) Ours | (f) GT |

Figure 7: Comparisons on iharmony4 dataset. From left to right are composite images, the results of DucoNet [44], CDTNet [5] , our results, and ground truth, respectively.

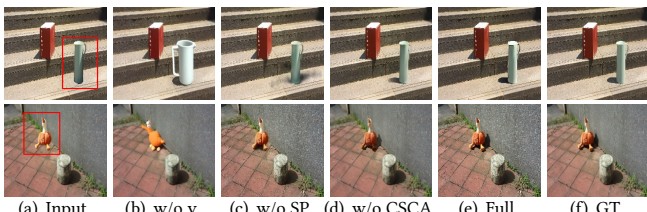

(a) Input    (b) w/o y    (c) w/o SP    (d) w/o CSCA    (e) Full    (f) GT

Figure 8: Ablation study results. The pictures fully demonstrate the effectiveness of image condition (y), coarse shadow predict module (SP) and channel-spatial cross-attention module (CSCA).

**Quantitative comparison.** Table 1 reports the comparison results on IH-SG test set. It can be observed that our method achieves the best quantitative results across all four evaluation metrics. This is mainly because existing image harmonization methods struggle to generalize well to outdoor real-world datasets, while existing shadow generation methods either rely on simple estimations of foreground shadow masks or directly generate shadows using learned data distributions. Such inaccurate estimations often lead to inferior results. In contrast, our method leverages the coarse shadow prediction module (SP) to effectively utilize background information, and the harmonization diffusion model can better guide the lighting editing of inserted objects, refine features, bridge the lighting gap between inserted objects and background environments. Additionally, by iteratively refining shadow regions, our method achieves more realistic shadow effects closer to real images.

**Visual comparison.** We provide some visual comparison results in Figure 6. It can be observed that our method not only achieves lighting variations across different scenes but also achieves the best visual effects of realistic shadows. Among these competing methods, for ARShadowGAN, it is difficult to edit object lighting, and the generated shadows are not accurate in shape and direction.

Table 1: Results of quantitative comparison on our testing set. "↑" indicates the higher the better, and "↓" indicates the lower the better. The best results are marked in bold.

| Method | RMSE ↓ | SSIM ↑ | fMSE ↓ | fSSIM ↑ |
|---|---|---|---|---|
| DucoNet [44] | 7.249 | 0.858 | 452.65 | 0.917 |
| DIH-GAN [1] | 6.108 | 0.849 | 579.12 | 0.886 |
| ObjectStitch [42] | 9.487 | 0.762 | 1249.48 | 0.794 |
| ARshadowGAN [25] | 9.146 | 0.812 | 977.81 | 0.807 |
| SGDiffusion [26] | 8.727 | 0.833 | 868.92 | 0.811 |
| Ours | **5.248** | **0.923** | **374.89** | **0.935** |

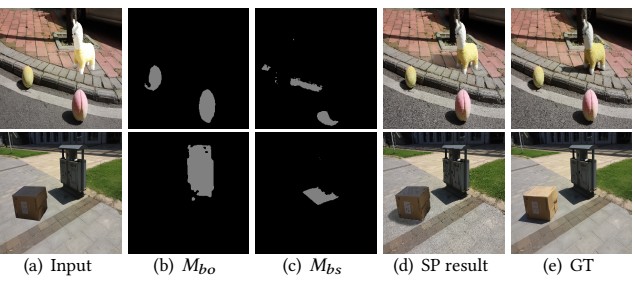

(a) Input    (b) $M_{bo}$    (c) $M_{bs}$    (d) SP result    (e) GT

Figure 9: Shadow prediction module. The second and third columns reflect the module's attention to background objects and their shadows.

On the other hand, SGDiffusion can generate relatively accurate shadows but still lacks in shape and shadow color accuracy. As for DucoNet, they fail to generalize well to outdoor real-world datasets. It aims to achieve visual harmony in images, which does not effectively address the problem of object shadows. The semantics of the image generated by the ObjectStitch have changed. In contrast, DIH-GAN, with its multi-scale attention mechanism and lighting feature exchange mechanism, can automatically infer object shadows and lighting generation. However, the shadows generated by this method lack completeness in details. In comparison, our model

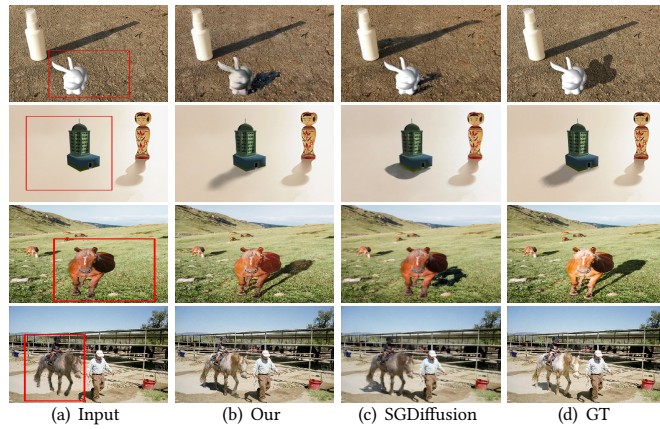

(a) Input  (b) Our  (c) SGDiffusion  (d) GT

**Figure 10: The results on the shadow generation dataset. The first two rows are images from the shadowAR dataset, and the last two rows are images from the DESOBAv2 dataset.**

can harmonize foreground objects and generate realistic and reasonable cast shadows.

### 5.4 Ablation Study

We study the impact of image condition y, shadow prediction module (SP), and channel-spatial cross-attention (CSCA) mechanism of our method on test images from IH-SH. The results are shown in Table 2, Figure 8.

 We visualized the coarse shadow predict module in Figure 9. It can be observed that model can effectively focus on the relevant areas in the background image, such as background objects and their shadows, and predict approximately correct shadows.

 To demonstrate the effectiveness of image conditions, we removed the guidance from images, denoted as "w/o y". The performance of "w/o y" is inferior compared to other models, indicating that utilizing image-condition guidance better preserves content structural information.

 To investigate the necessity of the coarse shadow prediction module SP, we removed this module, referred to as "w/o SP". It can be observed that without the SP module, there is a slight deficiency in shadow generation, and even the direction may be inaccurate. The performance of "w/o SP" is inferior to that of the full module, demonstrating the advantage of extracting background information and estimating coarse shadow regions.

 To demonstrate the effectiveness of the CSCA attention mechanism, the module was removed and replaced with a CAI layer [16], known as "w/o CSCA". The results are not as good as the entire model, indicating that this mechanism can help the model generate more realistic images.

### 5.5 Discussion

**Comparison on iHarmony4 [6] dataset.** The results in Figure 7 demonstrate the applicability of our method in image harmonization tasks. It can be observed that DucoNet [44] and CDTNet [5] do not effectively transfer low-level illumination to the foreground, while our method achieves the best results. Our method can bridge

**Table 2: Ablation experiments results. "↑" indicates the higher the better, and "↓" indicates the lower the better. The best results are marked in bold.**

| Method | RMSE ↓ | SSIM ↑ | fMSE ↓ | fSSIM ↑ |
|--------|--------|--------|--------|---------|
| w/o y | 9.372 | 0.669 | 1027.164 | 0.811 |
| w/o SP | 8.994 | 0.726 | 783.271 | 0.893 |
| w/o CSCA | 6.532 | 0.873 | 390.661 | 0.923 |
| Full | **5.524** | **0.915** | **362.713** | **0.935** |

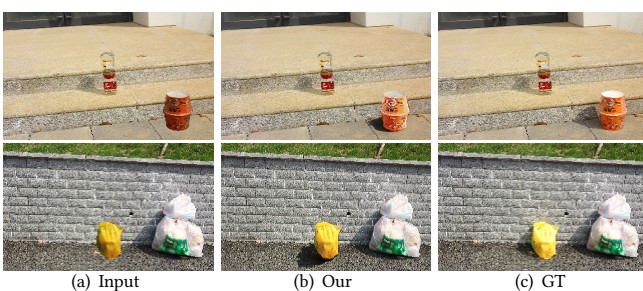

(a) Input  (b) Our  (c) GT

**Figure 11: Failed cases. There are difficulties in generating non planar cast shadows.**

the lighting gap between foreground objects and background environments while preserving the structural information of foreground objects, achieving lighting effects closer to ground truth (GT) images without changing the structure and detail of the foreground objects.

**Comparison on DESOBAv2 [26] and shadowAR [25] dataset.** We perturbed the foreground objects in the DESOBAv2. As depicted in Figure 11, the results demonstrate that our method can learn illumination information in the background to generate harmonious foreground objects and shadows for foreground objects. However, it is also observed that the generate shadows are somewhat unrealistic in few cases.

**Limitations:** As depicted in Figure 11, the proposed method has been successfully applied to image harmonization and shadow generation tasks in various environments. However, there are also some challenges. Firstly, our method faces challenges in generating non-planar cast shadows. This is because generating non-planar shadows requires more information, such as object geometry and environmental depth information, and the shadow generation results may be affected by mutual object occlusion.

## 6 CONCLUSION

In this work, we have introduced a diffusion model-based method to edit the lighting of foreground objects and generate visually reasonable cast shadows with preserving the structure of the image. In addition, we have proposed a large-scale high-quality outdoor real-world dataset IH-SG for image harmonization and shadow generation tasks. Our future work is to solve the generation of non-planar cast shadows of foreground objects.

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
