# OpenReview forum: "Foreground Harmonization and Shadow Generation for  Composite Image"
_acmmm.org/ACMMM/2024/Conference — MM2024 Poster_

### Official Review · Reviewer_P929 · 2024-05-24

**Rating:** 3
**Confidence:** 3

**Summary:**

This paper proposes a method for light and shadow editing of outdoor disharmonious composite images, including foreground harmonization and cast shadow generation. The proposed method can achieve controllable harmonization of foreground regions and a reasonable generation of cast shadows. Furthermore, this paper also proposes a new outdoor real-world dataset for image harmonization and shadow generation tasks. The experiment results show that the proposed method can generate shadows well and obtain good harmonization results.

**Strengths:**

1. This paper proposes a new outdoor real-world dataset for image harmonization and shadow generation, which is valuable for future research.
2. The paper is well-written and easy to understand.

**Limitations:**

1. The shapes of the shadows in the results presented in the paper do not closely match the shapes of the objects， such as Fig1.
2. The experiments and results presented in the paper are based on a resolution of 256x256. Can the model generalize to higher resolutions? Will the performance degrade at larger resolutions?
3. Introducing a new dataset should be one of the major contributions of this paper. However, there is no analysis of the dataset, such as data distribution and the overall size of the dataset. Why is this information missing?
4. I am curious about how the proposed method performs in complex cases, such as when the inserted objects overlap with background objects or when the background lacks clear shadow references.

**Suitability:**

2

---

### Official Review · Reviewer_yZ2i · 2024-05-24

**Rating:** 5
**Confidence:** 4

**Summary:**

The paper introduces an innovative approach to harmonizing light and generating shadows in outdoor composite images. Unlike prior works that tackle either foreground editing or shadow generation in isolation, this research addresses the interplay between lighting and shadow casting comprehensively. The authors leverage the robust capabilities of diffusion models, particularly their iterative denoising properties, to restore images while maintaining the content structure. A key contribution is the construction of the IH-SG dataset, which provides a diverse set of lighting conditions. The extensive experimental evaluation demonstrates the method's superiority over existing techniques.

**Strengths:**

1.The creation of a real-world outdoor dataset IH-SG that facilitates dual tasks of image harmonization and shadow generation. 2.A new image light-shadow editing method that achieves both controllable foreground harmonization and reasonable cast shadow generation. 3.Comprehensive experiments on public datasets and the newly created IH-SG dataset, which highlight the effectiveness of the proposed method.

**Limitations:**

1.Lack of further elaboration on dataset creation: more details on the dataset creation process, such as the variety of scenes, objects, and lighting conditions included, would be beneficial. Additionally, insights into any potential biases in the dataset and how they might impact the model's performance could be valuable. 2.How's the generalizability to diverse environments? The paper demonstrates impressive results on IH-SG dataset. However, it would be beneficial to understand how the model performs on a wider range of environments, particularly those with varied and non-uniform lighting conditions not represented in the current dataset. Could the authors provide insights or additional experiments on the model's robustness in such scenarios?

**Suitability:**

3

---

### Official Review · Reviewer_X1ir · 2024-05-24

**Rating:** 3
**Confidence:** 2

**Summary:**

This paper proposes a real outdoor dataset for image harmonization and shadow generation tasks. Propose a coarse shadow prediction network to predict cast shadows for foreground objects.Based on this dataset, it utilizes predicted prior with conditional diffusion model to generate higher quality shadows.

**Strengths:**

1.	The structure of this paper is clear and logical.
2.	This paper presents a new approach to shadow generation tasks and provides a new and practical dataset based on their method.
3.	The shadows generated by the method proposed in this article are better than those produced by existing methods.

**Limitations:**

1.	Based on the visual results presented in the paper, although the generated images are generally better than other methods, the overall visual quality of the results is not that good. Some high-frequency details are distorted and predicted shadows are somewhat unregular.
2.	While the network is clear, it appears rather simplistic. The application of the diffusion model seems somewhat conventional, and it's noticeable that the generated images lack texture details.
3.	The experiments are rather straightforward. Besides demonstrating the adaptability of their method, there is a lack of evidence showcasing the advantages of their proposed dataset. Additionally, there are few ablation experiments.

**Suitability:**

2

---

### Official Review · Reviewer_NQi8 · 2024-05-25

**Rating:** 2
**Confidence:** 4

**Summary:**

This paper addresses the problem of foreground light-shadow editing. Specifically, a coarse shadow prediction module (SP) is introduced to generate coarse shadows for foreground objects. Then, the predicted results as prior knowledge to guide the generation of harmony diffusion model. Moreover,  the authors construct a real outdoor dataset, IH-SG, covering various lighting conditions. Experiments on IH-SG demonstrate its effectiveness.

**Strengths:**

This paper constructs a real outdoor dataset, IH-SG for foreground light-shadow editing.

**Limitations:**

1.	The technical contribution of methodology is weak. The shadow prediction module is widely used in shadow generation, and the diffusion model has been adopted in image harmonization.
2.	The main experiment is only conducted on the constructed IH-SG. The authors only provide several visual results in section 5.5 on other datasets. However, the authors claim that extensive experiments conducted on both public datasets and our IH-SG dataset demonstrate the effectiveness of our method (lines 84-85, lines 181-182). It is overclaimed. The quantitative results on DESOBAv2 and DESOBAv2 are not provided.

**Suitability:**

2

---

### Official Review · Reviewer_aJda · 2024-05-26

**Rating:** 5
**Confidence:** 3

**Summary:**

The paper proposes a new method for light and shadow editing of outdoor composite images, aiming to achieve both foreground harmonization and realistic cast shadow generation. This is an important problem as lighting not only affects the brightness and color of objects but also produces corresponding shadows. Existing methods tend to focus either on foreground appearance editing or solely on shadow generation, while the paper addresses both aspects within one framework.
The proposed approach is implemented through two steps:  a coarse shadow prediction module to generate initial shadows for foreground objects and a refined module to generate high-quality cast shadow containing more details.

**Strengths:**

The paper deals with foreground appearance editing and  shadow generation tasks with in one framework. The paper proposes an integrated method that tackles both foreground harmonization and cast shadow generation in outdoor disharmonious composite images.

The paper utilizes diffusion models, which have demonstrated strong generative capabilities, especially in image restoration tasks. By leveraging the iterative denoising properties of diffusion models, the method can generate harmonious foreground images and realistic cast shadows.

A coarse shadow prediction module (SP) is used to generate coarse shadows for foreground objects. These predicted results serve as prior knowledge to guide the generation of the harmony diffusion model, resulting in improved shadow generation.

A new dataset called IH-SG is delivered in the paper, which enriches performance comparisons in the field.

**Limitations:**

The paper deals with foreground appearance editing and  shadow generation tasks with in one framework. But the paper does not provide details on the specific model architecture used, including the network design, loss functions, and training procedures. Without these details, it is challenging to replicate the method or analyze its strengths and weaknesses.

Some hyperparameters such as in equation (16)  are chosen emperically without giving any interpretation.

**Suitability:**

3

---

### Meta-Review · Area_Chair_jARW · 2024-06-26

**Recommendation:** Accept (Poster)
**Confidence:** 5

**Metareview:**

This paper receives a mixture of accepts and rejects. The negative conerns are mainly about insufficient experiments and flawed visual results. The authors provide the requested experimental results in the rebuttal, which verifies the effectiveness of the proposed method. Two reviewers mention that the generated shadows do not perfectly match the object shapes, and question the effectiveness in complex/occluded scenes. Since shadow generation is a very challenging task and there is still much room of improvement, the imperfect results cannot be solid reason for rejection. Considering the dataset contribution and effective method design, I tend to accept this paper.